# An Examination of the Variation in Estimates of E-Cigarette Prevalence among U.S. Adults

**DOI:** 10.3390/ijerph16173164

**Published:** 2019-08-30

**Authors:** David T. Levy, Zhe Yuan, Yameng Li, Darren Mays, Luz Maria Sanchez-Romero

**Affiliations:** Cancer Prevention and Control, Lombardi Comprehensive Cancer Center, Georgetown University, Washington, DC 20007, USA

**Keywords:** e-cigarettes, ENDS, vaping, prevalence, measurement

## Abstract

*Introduction*: Accurate estimates of e-cigarette use are needed to gauge its impact on public health. We compared the results of online and traditional, large scale surveys and provide additional estimates from the Population Assessment of Tobacco and Health (PATH) survey, with the aim of assessing the extent of variation in prevalence estimates. *Materials and Methods*: We searched the peer-reviewed literature for nationally representative estimates of U.S. adult e-cigarette prevalence, and developed our own estimates from waves one, two, and three of the PATH survey. We compared estimates by age, gender, cigarette smoking status, and e-cigarette use intensity both between online and traditional surveys and among the traditional surveys. Results: For specific years, online surveys generally yielded higher adult use rates than most traditional surveys, but considerable variation was found among traditional surveys. E-cigarette prevalence was greater for less intensive than for more intensive use. Levels of use were higher among current and recent former cigarette smokers than among former smokers of longer quit duration and never smokers, and by those of younger ages. *Conclusions:* Considerable variation in e-cigarette use estimates was observed even for a specific year. Further study is needed to uncover the source of variation in e-cigarette prevalence measures, with a view towards developing measures that best explain regular use and transitions between the use of e-cigarettes and other tobacco products.

## 1. Introduction

As the leading cause of preventable death [1], the prevalence of cigarette use (“smoking”) is commonly used to gauge potential societal health impacts and to set goals for reducing smoking [2,3,4]. An example is the U.S. Healthy People goal to reduce smoking prevalence to 12% by 2020 [5,6]. In addition, studies generally rely on changes in prevalence estimates to evaluate policy effectiveness [7]. While playing a central role in monitoring and evaluating tobacco use, studies have found discrepancies between estimates of U.S. cigarette prevalence in the large, traditional surveys [8,9,10,11,12,13]. Survey administration mode (telephone or in-person), respondent type (self or proxy), and survey response rate have been identified as potential reasons for the variations [11,12,14].

With the recent growth in e-cigarette use (“vaping”) [15,16,17], much attention has shifted to vaping [18,19,20,21,22,23,24,25]. Some have raised concerns about the potential of vaping to delay smoking cessation and contribute to smoking initiation, while others have argued that vaping is more likely to replace smoking [26]. Regardless, public health implications often depend on whether vaping is by current, former, or never smokers, and whether smokers who vape quit cigarettes [19,20,21,26]. 

All of the U.S. ongoing, established, and publicly available traditional surveys of tobacco use, except the National Smoking and Drug Use Survey, now ask about vaping. Online surveys also collect vaping information, and generally provide more up-to-date data on use and adoption. Despite being a common survey question, studies have not been conducted to compare vaping estimates between online and traditional surveys or to compare the variability of vaping estimates from traditional surveys. However, estimates of youth e-cigarette prevalence have been found to depend on the specific questions asked [27]. In addition, studies have observed that estimates of adult vaping [28] and youth vaping [29] depend on the specific measure adopted (e.g., ever vs. current use, less vs. more intense use). 

The purpose of this study was to compare published adult vaping prevalence estimates from the different U.S. nationally representative surveys. In addition, we provided our own estimates developed from the Food and Drug Administration-sponsored Population Assessment of Tobacco and Health (PATH) surveys. We compared use rates by age, gender, cigarette smoking status, and vaping intensity of use both between online and traditional surveys and among the traditional surveys. 

## 2. Materials and Methods 

### 2.1. Search Strategy

To collect estimates of e-cigarette prevalence, we conducted a search of online databases (e.g., PubMed, Google Scholar, EconLit, Web of Science, Social Science Research Network, and the Federal Trade Commission website). Our searches were confined to inclusion of at least one product term (e.g., “e-cigarette”, “ENDS”, or “vaping”) and at least one term related to use (e.g., “prevalence”, “use”, “someday”, or “every day”). The search was conducted through January 15 2019. Inclusion was limited to nationally representative studies in peer-reviewed journals, with one study omitted because it was not intended to be nationally representative [30].

With the relatively large number of studies, we limited this study to U.S. surveys in order to be able to make meaningful comparisons. We also limited the survey to studies that include adults, because youth studies often consider different age ranges and are generally less comparable. We considered current use (last 30 day, or every day-someday) rather than ever vaping since ever use can occur in previous years and increasingly represents past use. We included studies that estimate adult e-cigarette use, where use refers to any vaping product with or without nicotine. We also extracted e-cigarette use by age, gender, and cigarette smoking status (never, current, and former, including by years quit). Finally, we considered vaping intensity, distinguishing daily (“every day”) vs. nondaily (“someday”) use, and the number of days used in the past 30 days. We distinguished “traditional” from “online” surveys. Traditional surveys are considered those that are probability-based, ongoing, and that included data using face-to-face interviews or traditional mail. Online surveys can adopt convenience or probability sampling and primarily use a web survey but may use a mixed mode.

In view of the importance of the PATH survey for its size and longitudinal design, we included our own analysis of PATH as part of the assessed estimates. We replicated the results of two studies using PATH [31,32] and further distinguished e-cigarette use by gender, age, and smoking status in wave one (2013–2014), wave two (2014–2015), and wave three (2015–2016). Additionally, we categorized former cigarette smokers by quit years (<1, 1≤3, 3≤5, ≥5 years). The vaping products listed shifted from only “e-cigarettes” in wave one to both “ENDS (including e-cigarettes, e-cigar, e-hookah, and e-pipe)” and “e-cigarettes (as one sub-type of ENDS)” in wave two and three. For consistency, we used e-cigarettes for the analyses in all waves. We used the suggested pre-processed method from the PATH Codebook [33] for measuring daily/nondaily and past-30-day use, and used the self-response weights provided in PATH for the mean and confidence interval estimates.

To measure the percentage of past-30-day users that were at least 5- and 20-day e-cigarettes, we used the PATH question about the number of days vaping in the past 30 days for someday, experimental every day, and former users (in waves two and three) and assumed other daily users were “frequent (at least 20 days in past month)” users. Since waves one and two only asked about days use by e-cigarette users but no other ENDS users, our analyses for past-30-day estimates were restricted to e-cigarettes in waves one and two but included ENDS use in wave three. 

### 2.2. Comparison of Studies

Results extracted from the included papers and our analyses are presented as mean percentages and 95% confidence intervals (CI) when available. Descriptive comparisons were made between estimated means and/or 95% confidence intervals of the studies, focusing on variations in estimates between studies using online and traditional surveys, among studies using traditional surveys, and among studies using different measure of use (e.g., daily-nondaily vs. any past-30-day use). 

## 3. Results

### 3.1. Methods and Measures of E-Cigarette Use

Table 1 provides a summary of the survey methods and measures used in different studies. In addition to our estimates from PATH, we found 28 studies (*n* = 11 online and *n* = 17 traditional) with a variety of sampling methodologies. Traditional surveys used data from six national surveys: National Health Interview Survey (NHIS) [34,35,36,37,38,39], National Adult Tobacco Survey (NATS) [40,41,42,43], PATH [31,32,44,45], Tobacco Use Supplement of the Current Population Survey (CPS-TUS) [46], Behavioral Risk Factor Surveillance System (BRFSS) [47], and National Health and Nutrition Examination Survey (NHANES) [48]. Online surveys primarily employed the GfK/Knowledge online panel either on its own (*n* = 3) [49,50,51], by selecting a custom sample (*n* = 5) [15,52,53,54,55], or in combination with other data (*n* = 1) [56]. One online survey [57] recruited a household random sample using random digit dialing and administered via phone or the web. One survey used a national consumer-based web panel [58].

Measures based on questions about daily and non-daily use only were adopted in 12 studies [31,32,34,35,36,37,38,39,40,45,47,56]. Two studies using NATS [41,43] and two online studies [55,59] included “rarely use” along with daily and nondaily use. One study [44] distinguished daily and nondaily regular users from current triers who have not regularly vaped. Measures that included any past-30-day use were adopted in seven studies [15,46,49,51,52,53,58]. Another study [48] used any past-5-day prevalence. One study [32] defined current use as those who ever used “fairly regularly” and now use daily or nondaily, as well as those who used at least once in the past 30 days. One study [57] measured use as those who tried an e-cigarette and generally included those who currently use e-cigarettes, and another [50] considered daily, weekly, or monthly use.

Eight studies distinguished between e-cigarette intensity and duration of use. Two studies [34,35] provided daily vs. nondaily use, one study [44] distinguished regular use from triers and daily vs. nondaily use, and four studies distinguished rarely use [41,42,43,53]. One study [49] defined use as those who vaped at least once in the past 30 days and more than 50 times during their lifetime. Three studies [31,46,53] considered the number of days used in the past month.

### 3.2. All Adult Use

Using an adult daily-nondaily measure with online samples, e-cigarette use was at 0.3% in 2010 increasing to 6.8% in 2013 [56], at 7.4% in 2015 [55], and 8.5% in 2017 [53]. Traditional surveys using NATS reported a prevalence of 4.2%, with “rarely” use (1.9% without “rarely” use) for 2012 to 2013 [42], 5.4% in 2012 to 2014 [41], and at 6.6% in 2013 to 2014 (3.3% without “rarely” use) [43]. 2013 to 2014 PATH estimates were at 5.5% [31,32] and 5.6% [44]. We estimated PATH rates at 5.5% in 2013 to 2014, 5.3% in 2014 to 2015, and 4.4% in 2015 to 2016. NHIS studies reported use at 3.7% in 2014 [34,37,38], 3.5% in 2015 [38], 3.2% in 2016 [38], and 2.8% in 2017 [39]. BRFSS 2016 use was 4.5% [47].

Among the online survey studies defining adult vaping prevalence in terms of any past-30 day use, current use was estimated at 1.4% in 2012 [53], and at 4.8% [58] and 4.9% [52] in 2014. A multi-year study found use increasing from 1.3% in 2010 to 2011 to 1.9% in 2012 to 2013 [15]. Using traditional surveys, past-5-day use was 2.6% using 2013 to 2014 NHANES [48], and past-30-day use was 6.7% using 2013 to 2014 PATH [32], and 2.1% using 2014 to 2015 CPS-TUS [46]. Our PATH estimates had use at 6.7% in 2013 to 2014, 6.3% in 2014 to 2015, and 7.0% in 2015 to 2016.

As shown in Figure 1, large variations in adult vaping are seen even for the same time period, particularly for traditional surveys. Traditional surveys indicated rates with non-overlapping confidence intervals in the range of 2.6% (95% CI = 2.0%–3.1%) from 2013 to 2014 NHANES [48] and our estimate of 6.7% (95% CI = 6.4%–7.0%) from 2013 to 2014 PATH, of 2.1% (95% CI = 2.0%–2.1%) from 2014 to 2015 CPS-TUS [46] and 6.5% (95% CI = 6.2%–6.9%) from 2014 to 2015 PATH, and of 3.2% (95% CI = 2.9%–3.5%) from 2016 NHIS [38], and 4.5% (95% CI = 4.4%–4.6%) from 2016 BRFSS [47]. Online surveys showed higher rates over the years 2013 to 2017 than traditional surveys, with estimates of 6.8% (95% CI = 5.9%–7.7%) in 2013 [56], 4.9% (95%CI = 4.3%–5.5%) in 2014, and 7.4% (95% CI = 6.6%–8.4%) in 2015 [46] and 2017 (8.5%, CI not provided) [50]. In addition, online surveys indicate a generally increasing trend in vaping from 2010 to 2017, while trends appear to decline since 2014 slightly based on NHIS. Prevalence does not generally appear to systematically vary between surveys using past-30-day use vs. daily-nondaily measures, except when rarely used is included. However, PATH daily-nondaily measures are less than past-30-day measures. 

### 3.3. Gender and Age Variations for All Adults

Adult vaping rates from online samples were generally similar by gender, with differences of <1% percentage points [15,53,56]. Traditional surveys, including PATH estimates, yielded higher male than female rates with differences ≤3% percentage points [34,36,38,39,42,43,46,47,48].

Distinguishing by age group, online surveys reported past-30-day use as 0.9% for ages 18 to 24, 2.1% for ages 25 to 44, 2.3% for ages 45 to 64, and 1.6% for ages 65+ in 2012 to 2013 [15]. Similar age patterns were obtained for 2012 [53] using past-30-day use and for 2013 using daily-nondaily use [56]. The 2014 to 2015 PATH daily-nondaily prevalence was estimated at 13.2% for ages 18 to 24 and 2.5% for ages 25 and above [32]. For that same data but using past-30-day use, estimates were at 10.3% for ages 18 to 24, 7.4% for ages 25 to 44, 4.1% for ages 45 to 64, and 1.3% for ages 65+. Our 2014 to 15 PATH daily-nondaily use estimates were 13.4% for ages 18 to 24; 8.6% for ages 25 to 44; 4.8% for ages 45 to 64; and 1.5% for ages 65 and above. Similar decreasing trends with age were observed using 2013 to 2014 NHANES [48], 2014 to 2015 CPS-TUS [46], NATS [41,42,43], 2016 BRFSS [47], and NHIS [34,36,37,38,39].

### 3.4. Smoking Status

#### 3.4.1. Current cigarette smokers

Using an adult daily-nondaily vaping measure confined to current cigarette smokers, online survey studies estimated rates at 6.0% in 2010 to 2011 [57], 24.1% in 2014 [54], 29.8% in 2015 [55], and showed an increase from 1.4% in 2010 to 31.4% in 2013 [56]. For 2016, vaping rates were 10.6% among daily and 13.2% among nondaily smokers [50]. Traditional surveys had daily-nondaily use at 21.3% in 2013 to 2014, 21.7% in 2014 to 2015, and 14.4% in 2015 to 2016 using PATH, but other studies [34,37,38] reported use at 15.9% in 2014, and 14.9% [35] and 13.6% [38] in 2014 to 2015, and 10.8% [38] in 2016 using NHIS, and 14.4% using 2016 BRFSS [47]. One study [40] found more vaping among smokers who did vs. those who did not make a previous year quit attempt with 10.4% vs. 5.9% in 2012 to 2013 and with 14.8% vs. 10.7% in 2013 to 2014. 

Using a past-30-day measure, adult vaping increased from 4.9% in 2010 to 2011 to 9.4% in 2012 to 2013 [15], but other studies estimated use at 6.3% in 2012 [53], and 20.7% [52] and 20.6% [58] (without a lifetime 100 cigarettes screen) in 2014. Using PATH, we estimated smokers past-30-day vaping prevalence at 24.0% in 2013 to 2014, 22.1% in 2014 to 2015, and 20.2% in 2015 to 2016, but 2014 to 2015 CPS-TUS estimates were at 10.6% [46]. Past-5-day use was 8.2% from 2013 to 2014 NHANES [48]. 

As shown in Figure 2, online surveys after 2012 indicated higher rates than traditional surveys. For example, 20.7% (95% CI = 18.1%–23.6%) [52], 24.1% (CI not provided) [54] from online surveys are greater than 14.9% (CI not provided) [35] and 15.9% (95% CI = 14.2%–17.6%) [38] from traditional surveys in 2014. For 2015, an online survey had use at 29.8% (95% CI = 26.3%–33.5%) [55] compared to 13.6% (95% CI = 12.2%–14.9%) [38] from a traditional survey. Differences are also seen in estimates between traditional surveys, 8.2% (95% CI = 6.3%–10.1%) from NHANES [48] compared to 24.0% (95% CI = 23.1%–25.1%) from PATH in 2013 to 2014, 10.6% (95% CI = 10.2%–11.0%) from CPS-TUS [46] compared to 21.7% (95% CI = 20.6%–22.9%) from PATH in 2014 to 2015, and 10.8% (95% CI = 9.6%–12.0%) from NHIS [38] compared to 14.4% (95% CI = 13.9%–14.9%) from 2016 BRFSS [47]. PATH daily-nondaily measures are less than past-30-day measures. 

#### 3.4.2. Former smokers 

As shown in Figure 3, current adult daily-nondaily vaping among former smokers from online surveys increased from 0.3% in 2010 to 5.4% in 2013 [56], and 5.3% in 2015 [55]. Traditional surveys using PATH-estimated vaping at 3.9% in 2013 to 2014, 5.3% in 2014 to 2015, and 4.9% in 2015 to 2016. NHIS had use at 3.8% [34,38] (weighting former smokers by years quit) in 2014, 4.7% [38] in 2015, and 4.8% [38] in 2016, while 2016 BRFSS was 7.6% [47]. Past-30-day vaping from online surveys increased from 1.0% in 2010 to 2011 to 1.3% in 2012 to 2013 [15], and 4.0% [58] and 3.8% [52] in 2014. Traditional surveys yielded 2.7% from 2013 to 2014 NHANES [48] and 2.8% from 2014 to 2015 CPS-TUS [46]. Our PATH estimates increased from 4.9% in 2013 to 2014 to 6.3% in 2015 to 2016.

Differentiating by length of time quit (not shown in Figure 3), an online survey using a daily-nondaily measure yielded an adult vaping rate of 6.1% among those who quit ≤1 year and 0.2% among quitters >1 year in 2012 [53]. For 2015, use [55] was at 24.7% among those who quit <1 year, 16.5% among those who quit between 1 and 5 years, and 2.0% among >5 years quitters. In 2016, reported daily/weekly/monthly e-cigarette use was 14.3% among former smokers who quit ≤2 years [50]. Two 2014 NHIS studies [34,37] reported daily-nondaily vaping prevalence much lower for those <1 year quit than among those ≥1 year. Using past-30-day, 2013 to 2014 PATH had vaping at 27.9% for quit <1 year, 14.1% for quit 1 to 2 years, 7.3% for 3 to 5 years quit, and 1.2% for quit ≥5 years. 2014 to 2015 CPS-TUS estimates [46] were at 16.3% for quit <1 year, 9.8% for 2 to 3 years, and 0.5% for quit ≥3 years. The 2015 to 2016 PATH past-30-day use was at 21.8% for quit <1 year, 22.3% for 1 to 2 years, 28.7% for quit 3 to 4 years, and 8.5% for quit ≥5 years.

#### 3.4.3. Never Smokers

As shown in Figure 4, online surveys using a daily-nondaily adult vaping measure showed an increase from 0.1% in 2010 to 2012 to 1.4% in 2013 [56] and 3.0% in 2015 [55]. Past-30-day use was at 0.2% in 2010 to 2011 [15], 0.04% in 2012 [53], and 0.8% [58] and 0.9% [52] in 2014. Traditional surveys with daily-nondaily measures also showed an increase from 0.4% in 2014 [26] to 0.6% in 2015 [29] to 0.7% in 2016 using NHIS [38]. We estimated use at 1.4% in 2013 to 2014, 1.3% in 2014 to 2015, and 1.3% in 2015 to 2016 with PATH, and BRFSS had use at 1.4% in 2016 [47]. For past 30-day use, never smoker vaping rates were at 0.4% from 2013 to 2014 NHANES [48] and at 2.2% with PATH. For 2014 to 2015 CPS-TUS [46] indicated 0.2%, and our 2014 to 2015 PATH rates were at 1.9%. 

### 3.5. E-Cigarette Intensity of Use 

Most vaping studies distinguished intensity by daily vs. nondaily use. Among online surveys, the ratio of all adult daily to nondaily use was 16%/84% in 2012 [53] and for daily/(nondaily use+ rarely use) was 23%/77% in 2017 [59]. From traditional surveys, the 2014 to 2015 NHIS daily/nondaily ratio was 30%/70%, varying from 17%/83% among ages 18 to 24 to 43%/57% among ages 65+ [34]. Another 2014 to 2015 NHIS study [35] reported rates of 34%/66%. PATH use in 2013 to 2014 was 21%/79%, with rates similar by gender but increasing from 15%/85% among ages 18 to 24 to 34%/66% among ages 65+. 2014 to 2015 PATH rates were at 23%/77% in 2014 to 2015 and increased to 33%/67% in 2015 to 2016. Using other intensity distinctions, 2013 to 2014 PATH [44] rates were at 18% regular daily use, 25% regular nondaily use, and 57% triers. Using the same data, another study [31] found 21% daily, 37% moderate, and 42% infrequent (1–2 days) use. 2013 to 2014 NATS [41] e-cigarette rates were at 19% daily, 29% nondaily, and 52% rarely use. For 2016, the BRFSS daily/nondaily ratio using was 33%/67% [47].

According to smoking status, studies with daily/nondaily e-cigarette measure found higher rates among former than current smokers. An online survey had 2012 rates of daily/nondaily vaping at 11.5%/88.5% among smokers, 46%/54% among ≤1 year former smokers, 31%/69% for >1 year quitters, and 0%/100% among never smokers [53]. 2014 NHIS ratios were at 22%/78% among smokers, 72%/28% among former ≤1 year, 63%/37% among quit 2 to 3 years, and 25%/75% among quit ≥4 years and never smokers [34]. PATH daily/nondaily use ratios among former smokers were 62%/38% in 2013 to 2014; 59%/41% in 2014 to 2015; and 69%/31% in 2015 to 2016. Among never smokers, PATH ratios were 11%/89% in 2013 to 2014, 7%/93% in 2014 to 2015, and 12%/88% in 2015 to 2016. 2016 BRFSS rates were 22%/78% among smokers, 66%/34% among former smokers, and 14%/86% among never smokers [47].

The 5+/20+ days as a percent of past-30-day use measures for 2014 to 2015 CPS-TUS [46] were 75%/45% among all users, 69%/34% among smokers, 91%/74% among former smokers, and 32%/14% among never smokers. Our PATH rates were 62%/32% in 2013 to 2014 and 52%/31% in 2014 to 2015 for all users. Among smokers, 5+/20+ day PATH rates were 58%/25% in 2013 to 2014 and 50%/24% in 2014 to 2015. Former smokers had rates at 87%/70% in 2013 to 2014 and 78%/66% in 2014 to 2015. Never smokers showed the lowest rates of e-cigarette use at 55%/21% in 2013 to 2014, and 31%/ 13% in 2014 to 2015. PATH 2015 to 2016 had overall ENDS use rates at 47%/29% overall, 45%/24% among current smokers, 73%/63% among former smokers, and 30%/12% among never smokers.

Finally, established use measured as “used an e-cigarette at least once in the past 30 days and more than 50 times in their lifetime” among current and former (<5years) smokers accounted for 31.0% and 17.5% of current past-30-day use among males and females in 2013, with higher rates among ages 45 to 59 than other age groups [49]. A 2016 online survey [50] had the ratio of daily/(weekly and monthly) e-cigarette use at 4.7%/5.9% among daily smokers, 7.0%/6.2% among nondaily smokers, and 11.7%/2.6% among former smokers (quit ≤2 years).

## 4. Discussion

Our study of U.S. adult vaping prevalence shows an increasing trend since 2010 with a possible leveling off since 2014. Unfortunately, the high level of variability found in e-cigarette rates for individual years made the results unsuitable for accurately assessing trends and characterizing the magnitude of use. We detected vast differences in the methodological approaches to collect data not just between but also within surveys. 

Comparing online and traditional surveys conducted since 2013, higher estimates were generally observed by online surveys. Most of the online study samples came from the Knowledge Panel (GfK) (Table 1). Due to its sampling methods, GfK surveys, which are conducted close to one another, may re-survey the same individual, potentially leading to bias from repeated sampling. This potential increases for sub-populations, such as current and former cigarette smokers. 

We also found considerable variation in adult vaping estimates from traditional surveys even for a given year. In 2014 and 2015, three major surveys, the NHIS, CPS-TUS, and PATH, provided estimates of vaping prevalence. Estimates from PATH were generally twice or greater than CPS-TUS rates, with NHIS prevalence generally in between. PATH daily-nondaily measures were also less than past-30-day measures. 

Similar discrepancies for adult smoking prevalence have also been observed among studies using traditional surveys [8,9,10,11,12,13,60], but with far less deviation. The variation in adult vaping estimates appears to be closer to those observed for smokeless tobacco use [61]. The apparent lack of consistency may be attributed to the variation in survey methodologic approaches observed for data collection, i.e., survey mode, respondent type (self vs. proxy), survey length and timing of questions, response rate, and the specific wording of the questions asked [11,12,14]. For example, the response rates shown in Table 1 varied from 25% to 83%. Some of the surveys were specific to tobacco use (e.g., NATS, most online surveys), while others covered a broader array of topics (e.g., NHIS, CPS-TUS). Surveys were predominantly cross-sectional; only two (PATH and ITC) were longitudinal. The length of the questions also varied, with PATH asking a more extensive series of e-cigarette questions than other surveys. 

Of particular importance in gauging the e-cigarette use are the many varieties of products at any point in time and their changes over time [62]. With third-generation e-cigarettes already being sold by 2013 [63] and the term “juuling” becoming popular by 2018 [64,65], the terminology used to describe the different types of devices may play an important role [65,66,67,68]. The descriptions of the products also varied among major surveys. While NATS just asked about e-cigarette use in general, NHIS gave specific examples, such as “vape-pens, hookah-pens, e-hookahs, or e-vaporizers”; CPS-TUS [46] asked about “vape-pens, hookah-pens, e-hookahs, e-vaporizers, e-cigars, or e-pipes”; NHANES [48] asked about “e-cigarette use” and showed examples of in-hand cards; and two surveys [52,58] asked about any of the ENDS, including “electronic or e-cigarettes; electronic hookahs, hookah pens, or vape pens; or some other electronic vapor product, such as e-cigars or e-pipes”. PATH described, “E-cigarettes look like regular cigarettes, but are battery-powered and produce vapor instead of smoke. There are many types of e-cigarettes. Some common brands include NJOY, Blue and Smoking Everywhere” in its first wave. PATH then generalized to “electronic nicotine products, such as e-cigarettes, e-cigars, e-pipes, e-hookahs, personal vaporizers, vape pens and hookah pens” in the second wave and further added an open classification of “something else” in the third wave. 

The variation in estimates across surveys may also reflect the transient use of e-cigarettes. We considered the intensity of e-cigarette use, including daily vs. nondaily use and the number of days in the past month. In general, the prevalence was higher for measures that required fewer days use in the past month, included “rarely” and infrequent use, and used past-30-day compared to past-5-day use. We found a tendency toward more intensive use over time, and most intensive use by recent quitters followed by current smokers and less recent former smokers. Substantially lower levels of intensity were found among never smokers. 

Research examining how the frequency of use affects prevalence estimates is clearly needed. Other measures, such as total use, e.g., at least 50 times lifetime use, used fairly regularly, and the duration of e-cigarette use, should also be considered in the development of suitable measures [62]. However, different measures may be appropriate for different applications. Regular use and longer duration of use [26,69] are more relevant in gauging the health impact of vaping. Shorter-term, less-intensive use may be relevant in gauging transitions from vaping to cigarette use or vice-versa [26,29]. With additional waves of PATH and the ITC surveys, these transitions can be systematically explored, taking advantage of the longitudinal nature of these surveys. 

## 5. Conclusions

Proper characterization of vaping use and trends is urgently needed to develop effective regulatory strategies. With considerable variation in adult vaping prevalence estimates even for a specific year, it is not clear which measure or methodology should be adopted in future studies. Indeed, different measures may be appropriate for different purposes and different approaches may shed light on different issues. Further exploration of the different measures is needed to determine which best capture tobacco use transitions and which best assess long-term health use. Without better-defined and consistent measures of use, the ability to evaluate the public health impact associated with vaping and other tobacco product use is likely to be increasingly problematic.

## Figures and Tables

**Figure 1 ijerph-16-03164-f001:**
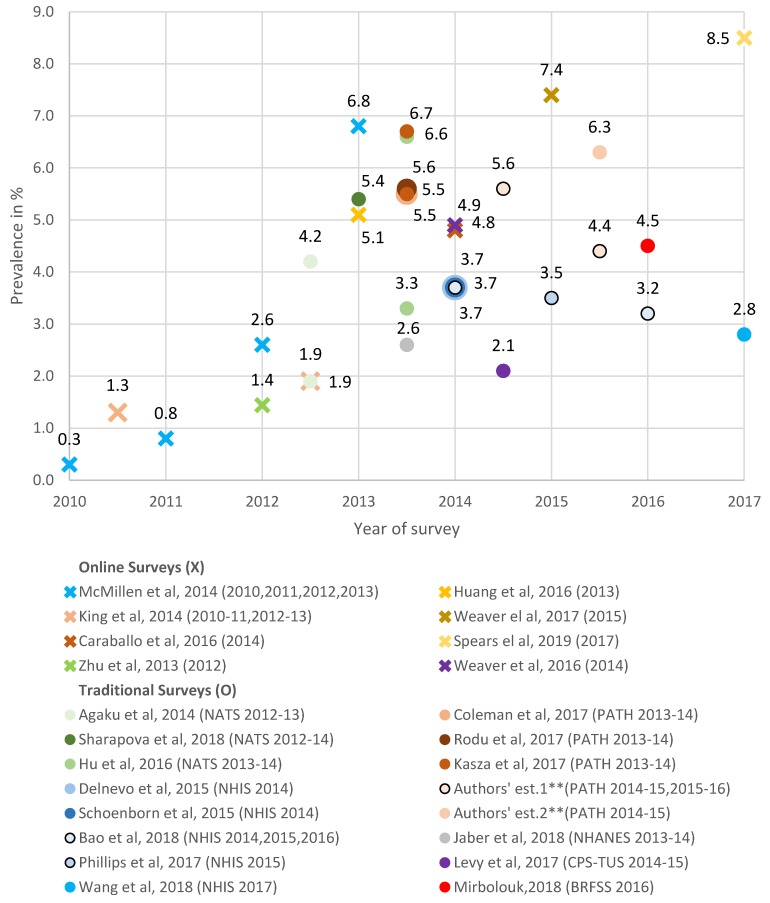
The prevalence of current e-cigarette/ENDS use among the total adult population, online and traditional survey estimates *, 2010–2017. * The estimates from online surveys are indicated by an X, and the estimates from traditional surveys are indicated by a circle. The legend adopts the format “author’s name, publication year (survey name and year(s)),” and the names of all online surveys are omitted. For surveys that collected data over a two-year period, we plotted the estimates in the mid-period (e.g., the estimate 2.6% from Jaber et al. using NHANES 2013–14 is plotted in 2013.5). ** Authors’ estimate one is measured by daily/non-daily ENDS use, and authors’ estimate two is measured by past-30-day ENDS use.

**Figure 2 ijerph-16-03164-f002:**
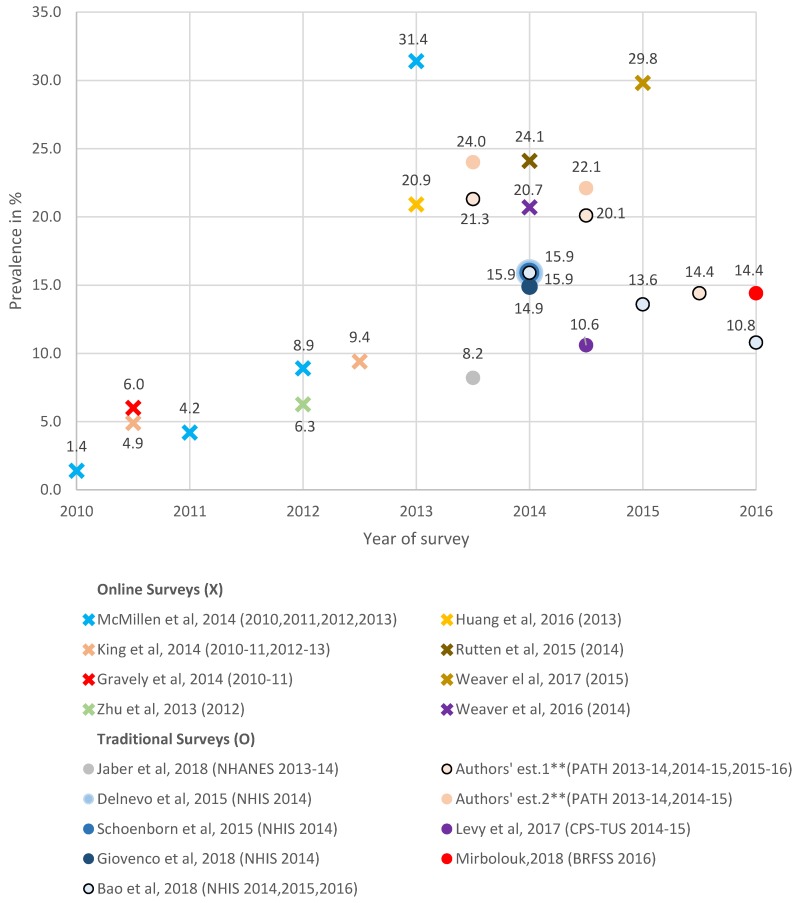
The prevalence of current e-cigarette/ENDS use among adult current smokers, online and traditional survey estimates *, 2010–2016. * The estimates from online surveys are indicated by an X, and the estimates from traditional surveys are indicated by a circle. The legend adopts the format “author’s name, publication year (survey name and year(s)),” and the names of all online surveys are omitted. For surveys that collected data over a two-year period, we plotted the estimates in the mid-period. ** Authors’ estimate one is measured by daily/nondaily ENDS use, and authors’ estimate two is measured by past-30-day ENDS use.

**Figure 3 ijerph-16-03164-f003:**
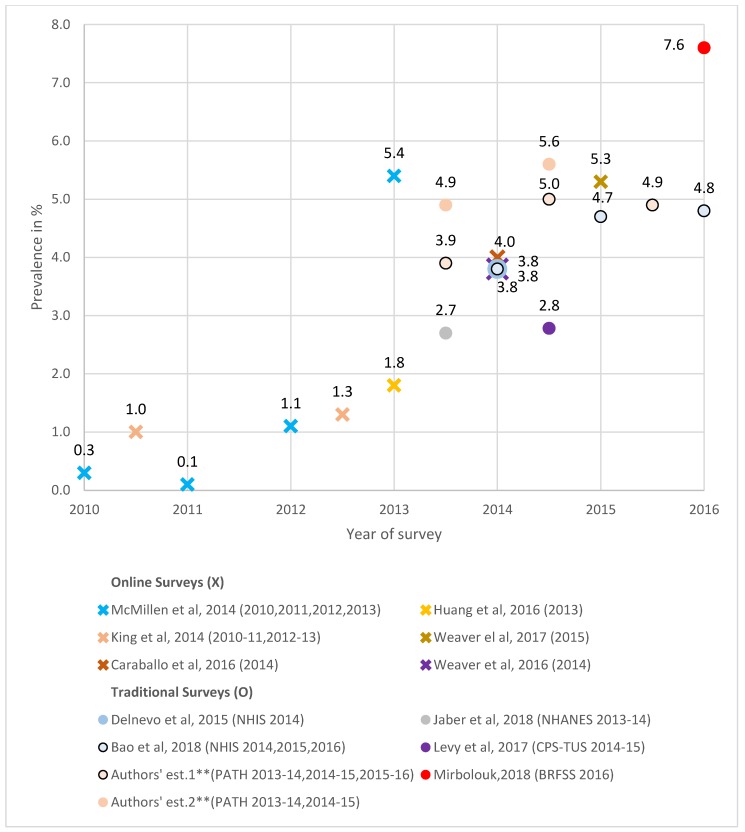
The prevalence of current e-cigarette/ENDS use among adult former smokers, online and traditional survey estimates, * 2010–2016. * The estimates from online surveys are indicated by an X, and the estimates from traditional surveys are indicated by a circle. The legend adopts the format “author’s name, publication year (survey name and year(s)),” and the names of all online surveys are omitted. For surveys that collected data over a two-year period, we plotted the estimates in the mid-period. ** Authors’ estimate one is measured by daily/nondaily ENDS use, and authors’ estimate two is measured by past-30-day ENDS use.

**Figure 4 ijerph-16-03164-f004:**
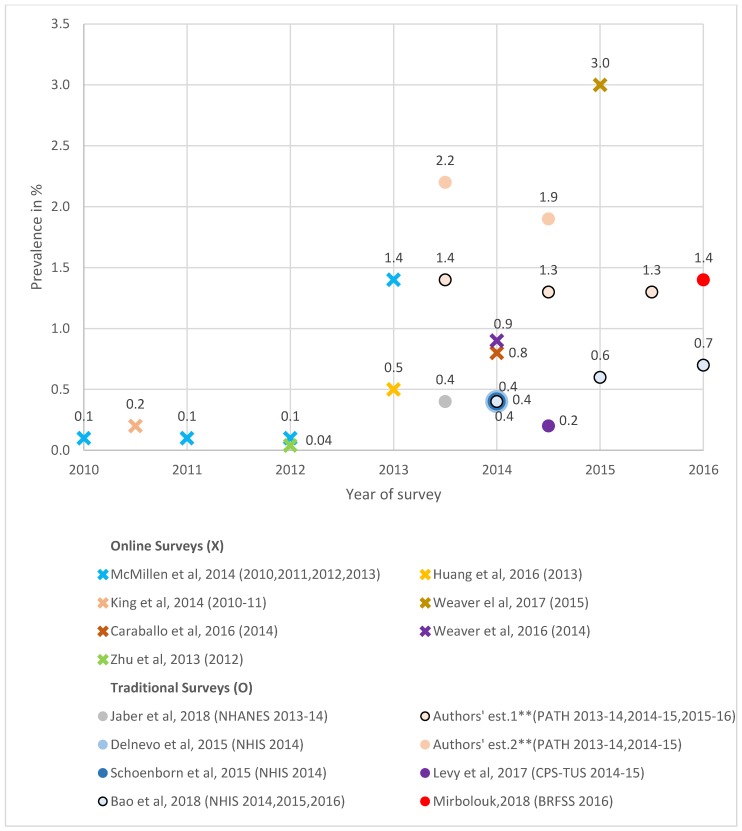
The prevalence of current e-cigarette/ENDS use among never smokers, online and traditional survey estimates *, 2010–2016. * The estimates from online surveys are indicated by an X, and the estimates from traditional surveys are indicated by a circle. The legend adopts the format “author’s name, publication year (survey name and year(s)),” and the names of all online surveys are omitted. For surveys that collected data over a two-year period, we plotted the estimates in the mid-period. ** Authors’ estimate one is measured by daily/nondaily ENDS use, and authors’ estimate two is measured by past-30-day ENDS use.

**Table 1 ijerph-16-03164-t001:** U.S. nationally representative surveys of current adult e-cigarette use.

Title	Survey Description (Name if Given/Dates)	Survey Description (Method, Surveyor)	Sample size (Response Rate/Completion Rate ^†^)	Current E-Cigarette/ENDS Use Definition	Other E-Cigarette Use Measurements
Online Surveys					
Zhu et al., 2013 [53]	February 24–March 8, 2012	Probability-based random digit dialing and address-based sampling by Knowledge Networks.	10,041 (66.5% ^†^)	Heard of and ever tried an e-cigarette; used e-cigarettes in the past 30 days.	Distinguished every day and some days use.
Giovenco et al., 2014 [49]	June 2013	Randomly recruited current and former smokers from Know-ledge Panel using probability-based sample from U.S.P.S. Delivery Sequence File.	2136 (NA)	Ever tried e-cigarettes; used e-cigarettes at least one day in the past-30 days.	Established users: current users who used more than 50 times over lifetime.
Gravely et al., 2014 [57]	July 2010–June 2011, International Tobacco Control (ITC) Survey	Probability sampling of smokers in households, random digit dialing, administered by phone or online.	1520 (25.6%) for U.S.	Heard of and tried an e-cigarette; currently use daily/less than daily, but ≥1 week/less than weekly, but ≥once a month/less than monthly.	NA
King et al., 2014 [15]	HealthStyles, June–August 2010–2013,	Probability-based random digit dialing and address-based sampling by Knowledge Networks.	2010 = 2505 (63.9%), 2011 = 4050 (69.1%), 2012 = 4170 (65.1%), 2013 = 4033 (66.1%)	In the past 30 days, used “electronic cigarettes or e-cigarettes” at least once.	NA
McMillen et al., 2014 [56]*	October–November 2010, 2011, 2012, and 2013	Probability-based random digit dialing and address-based sampling by Knowledge Networks.	2010 = 3240 (73.6% ^†^) 2011 = 3097 (65.1% ^†^) 2012 = 3101 (73.1% ^†^) 2013 = 3245 (74.5% ^†^)	Heard of and tried e-cigarettes; now use e-cigarettes every day or somedays.	NA
Rutten et al., 2015 [54]	Web-based panel survey of current cigarette smokers, April–May, 2014	Probability-based random digit dialing and address-based sampling by Knowledge Networks.	2663 (55.3% ^†^)	Now use e-cigarettes very day or somedays	NA
Caraballo et al., 2016 [58]	Styles, June–July 2014	Randomly recruited probability-based, address-based sampling to reach landline phones and internet users.	4269 (69%)	Any ENDS product use in the past-30 days.	NA
Huang et al., 2016 [51]	February–March 2013	Probability-based random digit dialing and address-based sampling by Knowledge Networks	13,144 (97%^†^) + 4378 tobacco users	In the past 30 days, used e-cigarettes every day, somedays or not at all.	NA
Weaver et al., 2016 [52]	Tobacco Products and Risk Perceptions Survey, June–November 2014	Probability sample drawn from KnowledgePanel, sampled via address-based sampling or random digit dialing.	5717 (74.4% ^†^; qualification rate 98.2 %)	Heard of and tried e-cigarettes; used at least once during the past-30 days.	NA
Weaver et al., 2017 [55]	Tobacco Products and Risk Perceptions Survey (TPRPS), Web-based, August–September 2015	Probability sample drawn from KnowledgePanel, sampled via address-based sampling or random digit dialing.	6051 (74.4% ^†^)	Current ENDS/ENNDS use; Ever used electronic vapor products and now used every day, somedays, or rarely.	NA
Gravely et al., 2019 [50]	International Tobacco Control (ITC), July–November 2016	Probability-based sampling frames for initial recruitment. Recruited via web-based KnowledgePanel, IPSOS and prior ITC panel	2552 (NA)	Ever tried an e-cigarette and currently (daily, weekly, or monthly) use.	Daily weekly and monthly e-cigarette use
Spears et al., 2019 [59]**	Tobacco Products and Risk Perceptions Survey August–September 2017	Probability sample drawn from KnowledgePanel, sampled via address-based sampling.	5992 (74.3% ^†^)	Ever used ENDS; now using ENDS every day, somedays or “rarely”	Every day ENDS use
Traditional Surveys					
Agaku et al., 2014 [42]	National Adult Tobacco Survey (NATS), 2012–2013	Stratified, random-digit-dialed landline (47.2%) and cellular telephone (36.3%) survey	27,026 (44.9%)	Ever used at least one e-cigarette; now use every day, somedays, or rarely	Every day/Someday/ Excluding rarely e-cigarette use.
Anic et al., 2018 [40]	National Adult Tobacco Survey (NATS), 2012–2013 and 2013–2014	Dual-frame Random Digit Dialing sample, with independent samples drawn from landline and cell phone frames	2012/13 = 8891 (NA), 2013/14 = 11,379 (NA)	Ever used an electronic cigarette one time in your life; now use every day or somedays.	NA
Sharapova et al., 2018 [41]	National Adult Tobacco Survey (NATS), 2012–2013 and 2013–2014	Dual-frame Random Digit Dialing sample, with independent samples drawn from landline and cell phone frames.	2012/13 = 60,192, (44.9%), 2013/14 = 75,233, (36.1%)	Now use every day, somedays, or rarely.	Every day/ some days/ rarely current use
Schoenborn et al., 2015 [43]	National Health Interview Survey (NHIS), 2014	A multi-stage area probability sampling design collected via personal household interviews	36,697 (NA)	Ever used an e-cigarette, and now use e-cigarettes every day or somedays.	NA
Delnevo et al., 2016 [34]	National Health Interview Survey (NHIS), 2014	A multi-stage area probability sampling design collected via personal household interviews	36,697 (NA)	Ever used an e-cigarette; now use every day or somedays.	Every day or someday e-cigarette use.
Hu et al., 2016 [35]	National Adult Tobacco Survey (NATS), 2013–2014	Stratified, random-digit-dialed landline (70%) and cellular telephone (30%) survey	75,233 (36.1%)	Ever used and used e-cigarettes every day, somedays or rarely.	Every day or someday e-cigarette use. With and w/out rarely use.
Phillips et al., 2017 [36]	National Health Interview Survey (NHIS), 2015	A multi-stage area probability sampling design collected via personal household interviews	33,627 (55.2%)	Used at least once; now use e-cigarettes every day or somedays.	NA
Giovenco et al., 2018 [35]	National Health Interview Survey (NHIS), 2014–2015	A multi-stage area probability sampling design collected via personal household interviews	15,532 (NA)	Ever used; currently use an e-cigarette daily or somedays.	Every day or someday e-cigarette use.
Wang et al., 2018 [39]	National Health Interview Survey (NHIS), 2017	A multi-stage area probability sampling design collected via personal household interviews	26,742 (53%)	Reported using electronic cigarettes at least once during lifetime; now used every day or somedays	NA
Bao et al., 2018 [38]	National Health Interview Survey (NHIS), 2014–2016	A multi-stage area probability sampling design collected via personal household interviews	2014 = 36,520, 2015 = 31,724, 2016 = 32,931, (response rate range = 68%–74%)	Ever used; now use -cigarettes every day or somedays	NA
Levy et al., 2017 [46]	Tobacco Use Suppl. of Current Population Survey (CPS-TUS), 2014–2015	NCI-sponsored survey with 64% of surveys completed by telephone and 36% completed in person.	163,920 (NA, 35% were eligible to answer tobacco use questions)	Ever used an e-cigarette even once; used at ≥1 day in last 30 days	Ever used e-cigarette and used at least 5, 10, and 20 days in past 30 days.
Jaber et al., 2018 [48]	National Health and Nutrition Examination Survey (NHANES), 2013–2014	A National Center for Health Statistics conducted survey via home-based interview.	5423 for adults (NA)	Used an e-cigarette in the previous 5 days	NA
Mirbolouk et al., 2018 [47]	Behavioral Risk Factor Surveillance System (BRFSS), 2016	Centers for Disease Control and Prevention and all states and participating territories of the United States collected data via landline and cellular telephone.	466,842 (NA)	Ever used e-cigarette or other electronic vaping products; now use every day or somedays considered current users	Everyday e-cigarette or other electronic vaping products use
Coleman et al., 2017 [31]	Population Assessment of Tobacco and Health Survey (PATH), September 2013–December 2014	In person longitudinal study using audio computer-assisted self-interview, with address-based, area-probability sampling, and in-person household screener.	32,320 (74%)	Have seen or heard, ever used, and now use e-cigarettes every day or somedays.	In-frequent users: 0–2 days; moderate users: use on >2 of days; and everyday use.
Kasza et al., 2017 [32]	Population Assessment of Tobacco and Health Survey (PATH), September. 2013–December.2014	In person longitudinal using audio computer-assisted self-interview, with address-based, area-probability sampling, and in-person household screener.	32,320 (74%)	Current use; now use every day or somedays.	Used in past 30 days; current regular use: ever used “fairly regularly,” now use every or some days.
Rodu et al., 2018 [44]	Population Assessment of Tobacco and Health Survey (PATH), September 2013–December 2014	In person longitudinal study using audio computer-assisted self-interview, with address-based, area-probability sampling, and in-person household screener.	32,320 (74%)	Aware of and ever use even one time; now use e-cigarettes every day or somedays	Current triers who have not used fairly regularly and regular users
Kasza et al., 2018 [37]	Population Assessment of Tobacco and Health (PATH) study, September 2013–December 2014 and October 2014–October 2015	In person longitudinal using audio computer-assisted self-interview, with address-based, area-probability sampling, and in-person household screener.	2013/14 = 34,235 (74%), 2014/15 = 26,439 (83.2%)	Ever used ENDS; now use ENDS every day or somedays***	NA
Our Analysis of PATH (2019)	Population Assessment of Tobacco and Health (PATH) survey, September 2013–December 2014 and October 2014–October 2015, and October 2015–October 2016	In person longitudinal using audio computer-assisted self-interview, with address-based, area-probability sampling and in-person household screener.	2013/14 = 32,320 (74%), 2014/15 = 28,362 (83.2%), 2015/16 = 28,148 (78.4%)	Current use: consider both “now use the product every day or some days” and “used e-cigarette at least once in past 30 days” of only e-cigarettes in 2013/14 and ENDS (including e-cigarette, e-cigar, e-hookah, and e-pipe) in 14/15 and 15/16.	Current regular/experimental use: ever used the product “fairly regularly”/not, now use e-cigarette every day/some days or at least 1 day in past-30 days.

NA = Not available. * This study contacted 2128 individuals with 1504 responses via RDD frame and contacted 2272 individuals with 1736 responses via web frame in 2010, from which we inferred a 73.6% (1504 + 1736)/(2128 + 2272) overall response rate. Similarly, we inferred 65.1% (1500/2282 via RDD and 1597/2476 via web) in 2011, 73.1% (1507/1713 via RDD and 1594/2529 via web) in 2012, and 74.5% (1552/1689 via RDD, 1693/2667 via web) in 2013. ^†^. Completion rates. ** We weighted this study’s estimates of ENDS use rates among “any mental health characteristics” and among “no mental health characteristics” participants by their respective prevalence. *** ENDS use alone was added to ENDS use with other products.

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
