# Peer review of "An Examination of the Variation in Estimates of E-Cigarette Prevalence among U.S. Adults"

_ijerph, 2019, doi:10.3390/ijerph16173164_

Round 1

Reviewer 1 Report

Overall, this paper addresses a critically important topic - understanding and contextualizing measurement and differences in nationally-representative prevalence estimates for product use in the US. The aims of this paper are ambitious - the authors compare estimates from 28 studies (as well as their own analyses) that use data from 6 "traditional" surveys and several online studies across 8 years. Overall, the paper is very well-done and provides an interesting contextualization of different prevalence estimates. However, there are several edits that could be made to improve the readability of the paper:

The "Materials and Methods" section may be easier to read with subsections to separate out the search strategy for included articles and the authors' own analyses Similarly, the authors do touch on the differences in different definitions of "current" use in the tables and later in the paper, however, it would benefit the reader to have some information about the differing definitions of current use in the methods. I believe that the authors are using "traditional surveys" to refer to established, publicly-available tobacco surveys, but this term a little confusing when contrasted with "online surveys." It's unclear if the authors intend to use "traditional" to just refer to "traditional" survey modes. I'd urge the authors to consider adding definitions or using different terms Throughout the paper, both "CPS-TUS" and "TUS-CPS" are used to refer to the Tobacco Use Supplement to the Current Population Survey – I’d suggest selecting one and using it consistently. Most commonly, I’ve seen “TUS-CPS” Page 5 of the document (Page 106 in the formatted PDF) appears to be blank The authors may want to consider re-organizing the “Traditional Surveys” section of the table so papers using the same survey are grouped together In the results, the authors do a good job of summarizing a significant amount of information – however, clarifying “study” vs “survey” and noting differences in definitions when comparing results On Page 12 (175 in the formatted manuscript) “PATH” is written as “PAT1H”

Author Response

Reviewer 1:

Overall, this paper addresses a critically important topic - understanding and contextualizing measurement and differences in nationally-representative prevalence estimates for product use in the US. The aims of this paper are ambitious - the authors compare estimates from 28 studies (as well as their own analyses) that use data from 6 "traditional" surveys and several online studies across 8 years. Overall, the paper is very well-done and provides an interesting contextualization of different prevalence estimates. However, there are several edits that could be made to improve the readability of the paper:

1)The "Materials and Methods" section may be easier to read with subsections to separate out the search strategy for included articles and the authors' own analyses Similarly, the authors do touch on the differences in different definitions of "current" use in the tables and later in the paper, however, it would benefit the reader to have some information about the differing definitions of current use in the methods.

Response: As suggested, we added subheads to the Methods section to distinguish Search Strategy, Our PATH estimates and Comparison of Estimates. We have added (last 30 day or daily-nondaily) after current use and the different measures of use are discussed in the section on our PATH estimates, and are better described in the second paragraph of the Results section.

2) I believe that the authors are using "traditional surveys" to refer to established, publicly-available tobacco surveys, but this term a little confusing when contrasted with "online surveys." It's unclear if the authors intend to use "traditional" to just refer to "traditional" survey modes. I'd urge the authors to consider adding definitions or using different terms.

Response: Online studies can use convenience or probability sampling, but those included in the review were all probability samples (Table 1). We updated the Methods section to indicate that online studies can use probability or convenience samples, and continue “online” to characterize the studies reviewed. We also define the terms traditional and online. We have added at the end of the second paragraph of the Methods section the following, “We distinguish “traditional” from “online” surveys. Traditional surveys are considered those that are probability-based, ongoing, and that included data using face-to-face interviews or traditional mail. Online surveys can adopt convenience or probability sampling and used primarily a web survey, but may use a mixed mode.”

3) Throughout the paper, both "CPS-TUS" and "TUS-CPS" are used to refer to the Tobacco Use Supplement to the Current Population Survey – I’d suggest selecting one and using it consistently. Most commonly, I’ve seen “TUS-CPS” Page 5 of the document (Page 106 in the formatted PDF) appears to be blank.

Response: We have change the use of “TUS-CPS” to “CPS-TUS” so that all references are to “CPS-TUS”

4) The authors may want to consider re-organizing the “Traditional Surveys” section of the table so papers using the same survey are grouped together. In the results, the authors do a good job of summarizing a significant amount of information – however, clarifying “study” vs “survey” and noting differences in definitions when comparing results

Response:  We have reorganized the table as suggested, including studies for the same survey together. We have also gone through the results section and carefully distinguished where the terms “study” and “survey” would be appropriate.

5) On Page 12 (175 in the formatted manuscript) “PATH” is written as “PAT1H” 

Response: Change made as suggested.

Reviewer 2 Report

The authors compared PATH Waves 1,2,3 derived e-cigarette use estimates among U.S. adults with those of published e-cigarette estimates from other studies, demonstrating variations across different measures and surveys. The authors did a commendable job of synthesizing findings from across XX studies. As a researcher in this field, I think this piece serve as a useful reference to those looking for an overview of how these measures continue to evolve over time, and vary by data source.

My primary comment/suggestion would be to temper the language around the Conclusions that “this study raises more questions than it answers. With considerable variation in vaping prevalence estimates even for a specific year, it is not clear which measure or methodology should be adopted in future studies.” As the authors noted in the Discussion, “different measures may be appropriate for different applications”, and I would agree. Although it is helpful to discern these differences in measures and estimates across surveys, I believe the utility of difference approaches lend their own value added. Due to variations in sampling and survey measures (many of which the authors indicated), publications on trends in e-cigarette use patterns typically are confined to a single data source to optimize. To my knowledge, it is generally understood that traditional surveys that rely on multistage probability sampling designs as the National Health Interview Survey, continue to remain the principal sources of nationally representative estimates for these behaviors; this is supplemented by findings from other studies such as PATH, which can also speak to longitudinal changes. Moreover, these traditional surveys have sample sizes that can be up to 10x larger than web-based panels, which rely on post-survey adjustments (e.g. per Census Current Population Survey estimates) to be representative.

Minor comment – please edit “TUS-CPS” for consistency; in some cases it is referred to as “CPS-TUS”, which may cause some confusion.

Lastly, for the figures – I encourage the authors to edit these further to facilitate ease of interpretation. Is it possible to include a legend or caption further detailing the use of colors and/or patterns? Datapoints are currently labeled as (lead author, year of publication), but some studies correspond to multiple data points of various years. For readers who may be unfamiliar with these studies, it may be particularly confusing as to why some datapoints are between axes/years. It would be very helpful to know how many datapoints each study provides, and for what years (up front), without having to only rely on the graphs. Finally, it may benefit readers to use the same color groups when referring to similar data sources (e.g. KnowledgePanel studies could all be a hue of blue).

Author Response

Reviewer 2:

The authors compared PATH Waves 1,2,3 derived e-cigarette use estimates among U.S. adults with those of published e-cigarette estimates from other studies, demonstrating variations across different measures and surveys. The authors did a commendable job of synthesizing findings from across XX studies. As a researcher in this field, I think this piece serve as a useful reference to those looking for an overview of how these measures continue to evolve over time, and vary by data source.

1) My primary comment/suggestion would be to temper the language around the Conclusions that “this study raises more questions than it answers. With considerable variation in vaping prevalence estimates even for a specific year, it is not clear which measure or methodology should be adopted in future studies.” As the authors noted in the Discussion, “different measures may be appropriate for different applications”, and I would agree. Although it is helpful to discern these differences in measures and estimates across surveys, I believe the utility of different approaches lend their own value added. Due to variations in sampling and survey measures (many of which the authors indicated), publications on trends in e-cigarette use patterns typically are confined to a single data source to optimize. To my knowledge, it is generally understood that traditional surveys that rely on multistage probability sampling designs as the National Health Interview Survey, continue to remain the principal sources of nationally representative estimates for these behaviors; this is supplemented by findings from other studies such as PATH, which can also speak to longitudinal changes. Moreover, these traditional surveys have sample sizes that can be up to 10x larger than web-based panels, which rely on post-survey adjustments (e.g. per Census Current Population Survey estimates) to be representative.

Response: We have tempered our conclusions in the last paragraph, emphasizing that different measures may be needed for different uses. We now state, “Proper characterization of vaping use and trends and its potential impact for public health are urgently needed to develop effective regulatory strategies. With considerable variation in adult vaping prevalence estimates even for a specific year, it is not clear which measure or methodology should be adopted in future studies. Indeed, different measures may be appropriate for different purposes and different approaches may shed light on different issues. Further exploration of the different measures is needed to determine which best capture tobacco use transitions and which best assess long-term health use. Without better defined and consistent measures of use, the ability to evaluate the public health impact associated with vaping and other tobacco product use is likely to be increasingly problematic.”

2) Minor comment – please edit “TUS-CPS” for consistency; in some cases, it is referred to as “CPS-TUS”, which may cause some confusion.

Response: Change made as suggested. We CPS-TUS through out.

3) Lastly, for the figures – I encourage the authors to edit these further to facilitate ease of interpretation. Is it possible to include a legend or caption further detailing the use of colors and/or patterns? Data points are currently labeled as (lead author, year of publication), but some studies correspond to multiple data points of various years. For readers who may be unfamiliar with these studies, it may be particularly confusing as to why some data points are between axes/years. It would be very helpful to know how many data points each study provides, and for what years (up front), without having to only rely on the graphs. Finally, it may benefit readers to use the same color groups when referring to similar data sources (e.g. KnowledgePanel studies could all be a hue of blue).

Response: We have attempted to make the figures more user friendly. We have changed the title of each figure to distinguish online and traditional surveys. We have added a footnote to explain that the circles represent traditional surveys and the x’s represent online surveys. We have also added the year of each survey so that the reader would know why some surveys are between years and we explain in the first footnote the practice regarding surveys that cover more than one year. Finally, we have attempted to make those studies using the same survey like hues, while at the same time being able to distinguish the different surveys. See also responses 7 and 8 to Reviewer 3 below.

Reviewer 3 Report

This study compares e-cigarette prevalence findings from articles that used nationally representative samples. Prevalence rates are used to direct funding and legislative efforts, thus, discrepancies among samples could lead to misappropriated efforts. The authors include details on the studies, including the survey method, response rate, and e e-cigarette definition used, which allows for comparing samples. The authors identified variation across articles, without a specific reason. The work is timely and important for researchers and practitioners. I have several comments and questions.

General

The introduction discusses youth e-cigarette use and uses discrepancies in youth rates to provide rationale for the study (e.g., page 2, lines 50 and 51). Thus, I was surprised to see only assessments among 18 and up were included. Why were youth studies not included? At minimum, the reason to exclude youth surveys should be acknowledged.

The operationalization of traditional and online is unclear. Please provide a definition of ‘traditional surveys.’ Are these government funded or ongoing studies? Similarly, how was ‘online’ operationalized – it appears this category also includes some phone surveys.

There are numerous minor typos throughout. For example, repeatedly ‘an’ instead of ‘a’ before daily-nondaily measure (e.g., page 11, line 131; page 13, line 176), and on page 12, line 162, the word ‘PATH’ has a ‘1’ in it.

Introduction

Page 2, Line 51. Is this referring to the relationship of vaping during youth/young adulthood and subsequent smoking among adults or generally vaping and smoking among different age groups?

Methods

Page 2, Line 60-62. I think this might be missing a couple words – do you mean that you reviewed articles that included at least one product and one use term, or that you searched databases using the terms? Also, to clarify, are these example terms used or were these the only terms used? If examples, please add ‘e.g.,’. If these were the only terms used, please clarify why these derivatives were selected.

Results

The results are somewhat unclear. For example, line 117 on page 11 states ‘daily-nondaily measures were used in twelve studies’. Line 125 states ‘two studies provided daily vs nondaily use’. Additionally, line 118 states ‘two studies… and two online surveys included rarely use’. Line 127 states ‘four studies distinguished rarely use’ but cites only 3 of those studies cited in line 118. Is this a different measure, and, if so, how is it different?

Figures

Figure 1. Are the colors and order of articles listed arbitrary? If possible, it’d be great to have a gradient for year and/or the articles organized by year/prevalence. Also, it might be more easily interpreted if the Data source and year are listed, with author/year in parenthesis. As currently written, it’s not super intuitive. The numbers overlap with the circles and other numbers; suggest manually placing them or resizing to avoid overlap.

Figure 2, 3, and 4. Again, colors appear somewhat arbitrary, with some articles differently colored than in Figure 1/2/3. Suggest being intentional with color choice.

Table

Page 1, Lines 37-38 and page 17 line 295 discuss administration mode and respondent type – these might be useful to include within the table. This would allow for additional discussion regarding whether there is variation in results based on these factors.

Author Response

Reviewer 3:

General

1) The introduction discusses youth e-cigarette use and uses discrepancies in youth rates to provide rationale for the study (e.g., page 2, lines 50 and 51). Thus, I was surprised to see only assessments among 18 and up were included. Why were youth studies not included? At minimum, the reason to exclude youth surveys should be acknowledged.

Response: We have added the second sentence to the second paragraph of the Methods section, “We also limited the survey to studies that include adults, since youth studies often consider different age ranges and are generally less comparable.”

2) The operationalization of traditional and online is unclear. Please provide a definition of ‘traditional surveys.’ Are these government funded or ongoing studies? Similarly, how was ‘online’ operationalized – it appears this category also includes some phone surveys.

Response: We added a definition of the terms traditional and online. We have added at the end of the second paragraph of the Methods section the following, “We distinguish “traditional” from “online” surveys. Traditional surveys are considered those that are probability-based, ongoing, and that included data using face-to-face interviews or traditional mail. Online surveys can adopt convenience or probability sampling and used primarily a web survey but may use a mixed mode.”

3) There are numerous minor typos throughout. For example, repeatedly ‘an’ instead of ‘a’ before daily-nondaily measure (e.g., page 11, line 131; page 13, line 176), and on page 12, line 162, the word ‘PATH’ has a ‘1’ in it.

Response: Changes made as suggested

Introduction

4) Page 2, Line 51. Is this referring to the relationship of vaping during youth/young adulthood and subsequent smoking among adults or generally vaping and smoking among different age groups?

Response: To be more precise and avoid ambiguity, we have limited the sentence just to measure of adult and youth vaping. We now state, “studies have observed that estimates of adult vaping [28] and youth vaping [29] depend on the specific measure adopted (e.g., ever vs. current use, less vs. more intense use).

Methods

5) Page 2, Line 60-62. I think this might be missing a couple words – do you mean that you reviewed articles that included at least one product and one use term, or that you searched databases using the terms? Also, to clarify, are these example terms used or were these the only terms used? If examples, please add ‘e.g.,’. If these were the only terms used, please clarify why these derivatives were selected.

Response: We mean that we used at least one from each of the two groups of terms, and these were the only terms used. We now state, “Our searches were confined to inclusion of at least one product term (e.g., “e-cigarette”, “ENDS”, or “vaping”) and at least one term related to use (e.g., “prevalence”, “use”, “someday”, or “every day”).”

Results

6) The results are somewhat unclear. For example, line 117 on page 11 states ‘daily-nondaily measures were used in twelve studies’. Line 125 states ‘two studies provided daily vs nondaily use’. Additionally, line 118 states ‘two studies… and two online surveys included rarely use’. Line 127 states ‘four studies distinguished rarely use’ but cites only 3 of those studies cited in line 118. Is this a different measure, and, if so, how is it different?

Response: We have attempted to clarify each of the points raised by the authors. We have re-written the paragraph to state, “Measures based on questions about daily and non-daily use only were adopted in twelve studies [31, 32, 34-40, 45, 47, 56]. Two studies using NATS [41, 43] and two online studies [55, 59] included “rarely use” along with daily and nondaily use. One study [44] distinguished daily and nondaily regular users from current triers who have not regularly vaped. Measures that included any past-30 day use were adopted in seven studies [15, 46, 49, 51-53, 58]. Another study [48] used any past-5 day prevalence. One study [32] defined current use as those who ever used “fairly regularly” and now use daily or nondaily, as well as those who used at least once in the past-30 days. One study [57] measured use as those who tried an e-cigarette and generally included those  who currently use e-cigarettes, and another [50] considered daily, weekly, or monthly use.” We have also added the fourth study to previous line 127.

Figures

7) Figure 1. Are the colors and order of articles listed arbitrary? If possible, it’d be great to have a gradient for year and/or the articles organized by year/prevalence. Also, it might be more easily interpreted if the Data source and year are listed, with author/year in parenthesis. As currently written, it’s not super intuitive. The numbers overlap with the circles and other numbers; suggest manually placing them or resizing to avoid overlap.

Response: We list the articles in terms of those using online samples first followed by traditional surveys. The articles are listed in terms of the first year of the sample. We have added the years of the survey to each articles, and explain in the footnotes the manner in which each article is listed. We have attempted to make sure that the numbers do not overlap with the circles/x/’s. See also comment 3 to Reviewer 2 above.

8) Figure 2, 3, and 4. Again, colors appear somewhat arbitrary, with some articles differently colored than in Figure 1/2/3. Suggest being intentional with color choice.

Response: The articles colors have been changed to match in all figures, and as described above (see comment 3 to Reviewer 2 above), we have attempted to make the colors less arbitrary.

Table

9) Page 1, Lines 37-38 and page 17 line 295 discuss administration mode and respondent type – these might be useful to include within the table. This would allow for additional discussion regarding whether there is variation in results based on these factors.

Response: Table 1 includes administrative mode (if we understand the reviewer correctly). We feel that respondent type is unnecessary, since all studies include only self-response.

Reviewer 4 Report

IJERPH-569042: An Examination of the Variation in Estimates of E-Cigarette Prevalence Among US Adults

This study described the variation in rates of e-cigarette use across different nationally representative samples of US adults, and compared rates by online versus traditional surveys and by age, gender, cigarette smoking status. The authors reported finding that online surveys appeared to report higher rates of e-cigarette use, in addition to variation across traditional surveys. They also reported finding that e-cigarette prevalence was higher with less intensive use, among cigarette smokers and in younger adults.

This paper addresses emerging methodological issues associated with monitoring rates of e-cigarettes use. Specifically, this paper highlights the inconsistency of the nationally representative samples currently available. The paper was well organized and the figures were helpful and creatively put together.

 I have just a couple comments/questions, which might help to clarify the manuscript and its conclusions.

Were the sampling weights applied in all the estimates presented, including the analyses of PATH data? If not, this may explain some of the discrepancies. Can the authors clarify throughout the manuscript and figure legends that these are estimates for adults? Can the authors provide a rationale for why they did not consider estimates in youth?

Author Response

Reviewer 4:

This study described the variation in rates of e-cigarette use across different nationally representative samples of US adults, and compared rates by online versus traditional surveys and by age, gender, cigarette smoking status. The authors reported finding that online surveys appeared to report higher rates of e-cigarette use, in addition to variation across traditional surveys. They also reported finding that e-cigarette prevalence was higher with less intensive use, among cigarette smokers and in younger adults.

This paper addresses emerging methodological issues associated with monitoring rates of e-cigarettes use. Specifically, this paper highlights the inconsistency of the nationally representative samples currently available. The paper was well organized and the figures were helpful and creatively put together.

I have just a couple comments/questions, which might help to clarify the manuscript and its conclusions.

1) Were the sampling weights applied in all the estimates presented, including the analyses of PATH data? If not, this may explain some of the discrepancies.

Response: At line 83, we have added that we added that we “used the weights provided in PATH for the mean and the confidence interval estimates.”

2) Can the authors clarify throughout the manuscript and figure legends that these are estimates for adults?

Response: Change made as suggested.

3) Can the authors provide a rationale for why they did not consider estimates in youth?

Response: We view youth studies as generally more heterogeneous and thus less comparable. Keeping to adults makes the point that estimates vary considerably even for a relatively homogeneous grouping. We have added the second sentence to the second paragraph of the Methods section, “We also limited the survey to studies that include adults, since youth studies often consider different age ranges and are generally less comparable.”